# Measuring Experience of Inpatient Child and Adolescent Mental Health Services (CAMHS)

**DOI:** 10.3390/ijerph20115940

**Published:** 2023-05-24

**Authors:** Jacob Clark, Euan MacLennan

**Affiliations:** Whittington NHS Trust, London N10 3HU, UK

**Keywords:** experience, feedback, CAMHS, inpatient, measure, quality, PREMS

## Abstract

There has been an important drive towards embedding feedback and experience data to improve health services in the UK. The current paper examines the gap in evidence and the lack of adequate measures of inpatient CAMHS experience. It presents the context of inpatient CAMHS and what factors influence care experience, before exploring the current practices for measuring experience and the implications for young people and families. The paper explores the dialectic that—given the nature of the work balancing risk and restrictions in inpatient CAMHS—it is essential that patient voice is at the centre of quality measures, and achieving this comes with a great complexity. The health needs of adolescents are unique, as are the interventions of psychiatric inpatient care, but current measures in routine use are often not developmentally adapted and lack validity. This paper looks to interdisciplinary theory and practice to consider what the application of a valid and meaningful measure of inpatient CAMHS experience might incorporate. It makes the case that the development of a measure of relational and moral experience of inpatient CAMHS would have significant implications for the quality of care and safety of adolescents during a period of acute crisis.

## 1. Introduction 

Approximately 3500 young people under 18 are admitted to inpatient Child and Adolescent Mental Health (CAMHS) tier 4 units across the UK each year [1]. Hospitals provide 24/7 care for young people with complex social, developmental, and emotional needs, often struggling with severe self-harm, suicidality, psychosis, and/or behaviours that present challenges. The level of acuity is high, and there is a constant requirement to balance the risk and liberty alongside young people and their families [2]. Such intensive treatment and high dysregulation are demanding. Inadequate management and supervision can lead to unsafe practices, elevated levels of restrictions, and poor formulation of risk. Revelations of abusive staff–patient relationships at the Edenfield Centre [3] have led to calls for an urgent review of psychiatric inpatient care, with the “control and choice” of service users expected to be central to this process [4]. The Care Quality Commission (CQC) had inspected the Edenfield Centre and deemed it overall to be “good” on measures of effectiveness, caring, responsive, and well-led [5], exposing the potential for poor or abusive psychiatric inpatient care to go unseen. 

To safeguard against such abuse, across the NHS there has been an emphasis on collaborative practices, including placing emphasis on patient and family experiences of care into service development [6]. There has been a focus on improving community services to reduce admissions and reduced length of hospital stays [7]. Alongside these markers of quality, there has been an increasing recognition of the value of healthcare experience as a key outcome [8]. Given the nature of the environment and treatment, there are few settings in which an emphasis on obtaining feedback could be more important than inpatient CAMHS. However, there is a significant gap in both evidence and practice. Measures of ‘satisfaction’ that were created and validated in outpatient settings failed to examine the key factors known to shape service users’ inpatient experience. Furthermore, the unique developmental needs of adolescents have been neglected, with practices translated from samples of children or adults. A recent National Institute for Health Research review of the experiences of young people with a diagnosis of autism spectrum condition (ASC) and/or learning disability in inpatient CAMHS concluded that, “understanding the key influences on a young person’s experience is the foundation of any improvement strategy” [9]. In simple terms, those who experience the care hold unique knowledge about what needs improving. There are multiple reasons why no clear methodology or measure has been developed. With the challenge being great and varied, how does one assess the range of complex feelings about an experience of care that may have involved the Mental Health Act to keep you safe, locked doors, close observations, separation from family, relationships with a more professionals and periods of severe distress and dysregulation? In addition to this, as any parent may know—asking teenagers for feedback can be complicated. 

The following paper will argue that quality in inpatient CAMHS should be centred around capturing and embedding the measurements of young people. It will start by examining what factors influence the experiences of inpatient settings, before exploring current practices in obtaining patient feedback, and the specific challenges of measurement in adolescent inpatient services. The current paper primarily examines the experiences for young people admitted to psychiatric wards. The role of parents and carers in this process was also considered, however their experience merits close attention, which is beyond the scope of this paper. Finally, this paper will draw on Liebling’s work measuring the ‘social and moral climate’ of institutional settings [10]. It will argue that moving away from service satisfaction towards assessing the moral and relational experiences of inpatient CAMHS has greater validity and could be a valuable direction for improving the safety and outcomes of our young people. 

## 2. What Are the Experiences of Service Users and Carers in Psychiatric Inpatient Care?

The quality of inpatient services is a primary concern of mental health service users. Psychiatric admissions involve elevated levels of dysregulation, with interventions that require restrictions. Most of the literature published over the past 20 years into patient experience of inpatient care has been associated with investigative reports following abusive practices [11,12]. This focus continues with the government recently launching a rapid review of the safety of inpatient care following several national scandals [13]. 

The research literature has primarily explored experiences of adult inpatient units. The systematic review conducted by Wood and Alsway (2015) of the qualitative research into experiences of psychiatric inpatient identified eleven studies with three main themes [14]. The first theme, collaborative and inclusive care pertained to how much service users felt actively involved in their care, and the negative effects of having their viewpoint disregarded or feeling coerced in treatment. Collaboration related to for example their transitions home, service users associated relapse with a lack of involvement in planning, and their length of stay not matching their needs. The second theme pertained to the role of positive relationships, which included both negative and positive experiences of peers and staff influencing their coping, self-esteem, and recovery. Staff shortages, lack of privacy, and feeling dismissed all contributed to feelings of isolation, frustration, and low mood. The final theme was safe and therapeutic hospital environments, meaning whether the environment felt homely or welcoming, as well as disruptions on the ward such as other patients’ distress. A recent systematic review identified 72 studies examining in-patient experience, 24 of them having been conducted in the UK, with four themes found: the importance of high-quality relationships; averting negative experiences of coercion; a healthy and safe and enabling physical environment/ward milieu; and finally authentic experiences of patient-centred care. The role of staff was central to each theme. The authors concluded that there was “little high-quality evidence on what is important to patients” [15] (p. 329). In each of the above systematic reviews, children and adolescents were excluded, as were the parents and carers. 

## 3. Experiences of Inpatient CAMHS

The lived experience of adolescents in inpatient CAMHS remains relatively unexamined. Several reports have focused on specific components of young people’s inpatient CAMHS experience. Waldegrave and Roffe (2020) were tasked by the children’s commissioner to investigate how young people’s experiences differed when they were admitted informally compared with a detention under the Mental Health Act [12]. The children’s advocacy charity Article 39 estimate that three-quarters of young people in inpatient CAMHS are there informally i.e., they and/or their parents have consented to the admission [1]. The views of young people at four CAMHS inpatient units at a general adolescent unit (GAU), a psychiatric intensive care unit (PICU), and a specialist eating disorder unit all highlighted the impact of restrictive practices on the experiences of care, notably access to leave off the unit, and the impact of experiencing or witnessing restraint [12]. The accounts provided by these young people show the importance of having a developmental lens in understanding adolescent experience of psychiatric inpatient settings, for example: “three massive people showing up at my door”, “I’m a kid at the end of the day... it’s like a punishment... but, for being ill”, and “I was 15 so I didn’t get much of a say” (pp. 3–5).

A recent NIHR themed review (2021) explored the experiences of young people with autism spectrum condition and/or learning disabilities in inpatient CAMHS [9], a significant and increasing proportion of the inpatient CAMHS population who are more likely to be detained under a Mental Health Act [1]. In examining the literature and consulting to young people, families, and staff, four familiar themes were established as key to experience: the quality of the relationships, the use of restrictive practices, the normality of experiences, and good clinical outcomes. Each factor was interdependent and necessary to ensure a good experience of care [8]. Hartley, Redmond, and Berry (2022) interviewed eight young people, eight family members, and eight nursing staff specifically regarding their experiences of therapeutic relationships in inpatient CAMHS, concluding that “they are the treatment” [16] (p. 19). 

No known recent studies have examined the overall experiences of young people and parents/carers in inpatient CAMHS. The research that is available has used in-depth qualitative analysis of small samples with specific research questions, as there are no large-scale quantifiable data on the overall adolescent inpatient CAMHS experience. Themes from the research above are consistent with the adult inpatient experience, but there are significant developmental differences in adolescent admissions: the role of peer relationships, experience of rules, involvement of parents, phone use, social media, and ongoing education. The NIHR review concluded that there was little evidence on the views of young people, and that future work is needed to “capture feedback in real-time (and) to identify approaches most appropriate for CAMHS inpatient setting” [9]. The following section will outline ways in which feedback has become integral in improving experience in health care, how CAMHS has led advances, and the application to inpatient CAMHS. 

## 4. Use and Usefulness of Feedback

Multiple terms have been used to describe the process of obtaining service user views on care: satisfaction, engagement, perceptions, perspectives, experiences, views, and voice [16]. An NIHR review surmised that there were at least 38 different types of patient experience data, with a wide variety of uses: to inform commissioning decisions, benchmark, support patient choice, compliance with standards, redesign services, and frame care as patient-centred rather than outcome focused [8]. While there is consensus that feedback can provide value, current use and usefulness are limited. Feedback systems are often tied to imposed data compliance or associated with ‘putting out fires’ in complaint processes [17]; when primarily connected to corporate or managerial processes, practitioners are disconnected from the feedback process. Patient experience data are rarely incorporated alongside quality indicators such as safety and clinical effectiveness [18]. Arguably, community CAMHS has been at the forefront of advancing the use of patient feedback and measures of experience to inform routine practice and service development. 

## 5. Feedback in CAMHS 

The case for integrating feedback loops for young people and parents/carers in CAMHS has been well established [19]. Improving ways of obtaining and responding to feedback has arguably been the central tenet across practice developments in CAMHS in both routine clinical work and overall service design. The move to embed feedback in therapy developed in recognition so that the therapeutic relationship is central to improving outcomes, and session-by-session tools could enhance epistemic trust, reduce high treatment dropout, and address low agreement on the target problem between parents, children, and practitioners [20]. Goal-based outcomes place at the highest importance an exploration of what service users want [21].

Alongside the use of feedback tools in routine clinical practice, commissioning contracts have stipulated that user experience is reported at the end of contact with services. Despite this, just 14% of patients discharged from community CAMHS provide feedback about their overall experience [22]. Solberg, Larsson, and Jozefiak (2015) examined satisfaction questionnaires of 120 adolescent-parent pairs in CAMHS outpatients over 3–4 years, and the relationship with treatment outcomes [23]. They found good overall satisfaction on their consumer satisfaction questionnaire, with low-to-moderate correlation between parent and adolescent satisfaction. Pairings in which the adolescent had higher ‘Total Problems’ [24] were less satisfied with CAMHS. They did not find an association between adolescent or parental satisfaction with treatment outcome, highlighting the importance of satisfaction measures in complementing the indicators of symptom reduction outcomes. Parental satisfaction with CAMHS has been found to be slightly higher than children, more reliable, and recommended for use in benchmarking services [25].

Kennedy’s (2010) review of children’s services following perceived failings in performance concluded, “there should be only one indicator or criterion of successful performance: satisfaction with the service” [26] (p. 86). While there have been advances, mechanisms for promoting service users’ experience within CAMHS inpatient settings represent “one of the more challenging areas in the NHS to obtain meaningful feedback” [27] (p. 2). 

## 6. Measuring the Experience of Psychiatric Inpatient Care 

Given the nature of the work in psychiatric inpatient care, routine patient experience feedback should be central to care, yet there is a gap in both evidence and practice. A recent large EURIPIDES multimethod review of how patient experience data has been embedded across 57 NHS providers of inpatient mental health care found that “patient experience is poorly integrated, often conducted primarily for regulatory compliance” [27]. They found contributing factors including inadequate processes for experience data, poor operationalisation of the concept of patient experience, and tools not adapted for use in these settings [27]. Adolescents were excluded from the major EURIPES review. 

Multiple measures have attempted to capture satisfaction with adult inpatient services, but there is large variation in methods used, content, and validation [28]. Evans et al. (2012) co-constructed a 19-item measure of inpatient satisfaction with service users, Views on Inpatient Care (VOICE) with good psychometric properties [29]. VOICE addressed ward-related experiences including relationships with staff, and involvement in care and activities. However, it does not elicit multiple key factors described in the literature—ward milieu, peer relationships, family involvement, authenticity of care, normality of experiences, experiences of coercion, and other restrictive practices. Furthermore, each of these factors will be developmentally distinct for adolescents—while the VOICE measure was developed for an adult inpatient population. The care needs of adolescents are unique, and yet the adaptations implemented are so often inadequate [30]. The following section will outline the relational and technical challenges of measuring inpatient experience, with reference to the current tools being used. 

## 7. Measuring Experience of Inpatient CAMHS

There are no well-established measures of inpatient CAMHS experience. The best-known and most widely used assessment of service satisfaction is the friends and family test. This simply framed question brings to light the sensitivity of assessing satisfaction in inpatient CAMHS—it is hard to imagine recommending a psychiatric inpatient admission to a friend or family member, and even harder still when one considers the distinct developmental dynamics of adolescents’ family and peer relationships. A version of this question is one of twelve items on the Experience of Service Questionnaire (ESQ) [25], which is the most widely used measure of inpatient CAMHS experience in the UK. 

Implementation of the ESQ with young people and parents/carers on discharge is a quality standard for the Royal College of Psychiatry Quality Network for Inpatient CAMHS (QNIC). It is a freely available brief measure which assesses two constructs—satisfaction with care and satisfaction with environment, with three additional open text questions. The ESQ has demonstrated a good sensitivity, and encompasses robust psychometric properties in overall ‘Satisfaction with Care’, and therefore has been promoted for use as a service evaluation tool to benchmark services—particularly using parent reports. The ESQ was validated for its utilisation in community CAMHS settings. The validation study analysed ESQ feedback from 7067 CAMHS service users where there was a feedback form from parents/carers or a child. The sample from member services of CAMHS Research Outcome Consortium (CORC) across England and Scotland was predominantly white (77%), with the mean age of the children assessed being 11.2 years (SD = 3.6). 

There are clear limitations for its application to adolescent inpatient settings. Several items have a poor face validity in an inpatient environment: item 8, the facilities here are comfortable (e.g., waiting area); item 9, appointments at a convenient time; item 10, it is quite easy to get to the place I have appointments. Items examining satisfaction with care appeared more salient (e.g., items 1–7: people listened, were easy to talk to people, treated well, taken seriously, and know how to help), but there is an underlying issue that during an inpatient admission, young people will directly encounter many more professionals than in community interventions including large nursing, teaching and medical teams, and bank staff, and experience direct contact and have meaningful relationships with a bigger multi-disciplinary team, therefore it is difficult to know who a young person is expected to hold in mind when responding to satisfaction with care. Furthermore, the ‘halo effect’ during an inpatient admission is likely to play a significant role in responses i.e., the overall feeling associated with care will influence all reported experiences, and therefore any range of mixed experiences are subsequently lost, creating a rating bias as a result. Brown, Ford, Deighton, and Wolpert (2014) found in their validation of the ESQ that conceptually independent environmental factors and care-related factors were highly correlated (p. 433). They argue that such ‘halo effects’ capture a valid reflection of overall satisfaction; however, this effect is likely to present differently when assessing the satisfaction with care in the context of negative overtones associated with severe distress and dysregulation, painful attachment separation, and restrictive practices, including possible legal frameworks as part of care. The authors found children’s responses to be less reliable than parents, pointing to possible effects of “cognitive immaturity” (p. 444). In adolescent inpatient settings, not only was the average age higher compared with community CAMHS, but there is also a more marked difference between the care received by a parent and their child who may be admitted 24/7 and entirely absorbed in the care. Therefore, there is an even greater moral imperative to find a way to capture the experiences of both young people and their parents/carers. The parent/carer experiences are also unique to having a child in an inpatient admission: involvement in daily decisions about risk and restrictions, the experience of making telephone contact with the ward, confidence in their child’s safety, and opportunities for contact with their child including on the ward or on planned leave [31]. While the ESQ has been adopted by QNIC as a key monitoring tool for inpatient services across the UK, to the best of our knowledge there are no published data presenting the response rates, content of feedback, or different predictors of positive or negative experiences of care in inpatient settings. This evidence gap reflects the lack understanding of current uses of the ESQ, but also suggests its limited utility in these settings. 

Adapted versions of the CAMHS satisfaction scales have been attempted, but amendments are minimal and secondary to the initial scale validation process, with an insufficient number of participants with inpatient experience included [32]. 

The following section will draw on advances in the prison and forensic sectors to argue that we should move away from attempts to measure ‘satisfaction’ and instead towards an evaluation of the ward ‘climate’ to better assess the inpatient CAMHS experience. 

## 8. Social Climate: Moral and Relational Performance of Institutions

Liebling’s [33,34,35,36] work on the social and moral climates of prisons offers an important conceptual framework for assessing the quality of experiences in institutional settings. Liebling’s approach emerged from a 12-prison study of suicide which found that mean levels of distress in prisoners (with the levels of distress according to the General Health Questionnaire, GHQ-12) were significantly associated with the suicide rates of each prison. When controlling for prior vulnerability factors, they found that the ‘quality of life’ of an establishment was found to contribute to the levels of distress among the prisoners [33]. Liebling and colleagues have developed an intensive methodology termed the Measuring the Quality of Prison Life (MQPL+) survey, involving detailed observations, an appreciative inquiry methodology aimed at discovering from staff and prisoner’s what matters most to them, and what a good day might look like within that institution. The environment was evaluated across seven quality-of-life dimensions related to the social climate, conducted by a research team totalling at least 70 person days. Liebling and colleagues have found that there are stark differences between the institutional moral and relational experiences including respect, dignity, humanity, autonomy, and fairness, as well as safety and family contact, and that these are a matter of life and death, with “some environments are more survivable than others” [34] (p. 535). Liebling and Arnold’s 2004 study identified four dimensions of social climate which directly predicted suicide—perceptions of safety, personal development, family contact, and dignity [10]. Furthermore, frustration, poor relationships with staff, low levels of support on entry to prison, low levels of individual care, and lack of activities aimed at personal development were all found to be significantly correlated with distress (see Liebling and Ludlow, 2016 for summary, [35]). To date, Liebling and colleagues have analysed data from a survey of 224 prisons over five years, with a total of 24,508 prisoners completing their surveys. Such robust research findings should be instrumental in attempts to redirect suicide prevention work towards the promotion of well-being and improving the quality of life at these institutions. 

Liebling argues for a move away from relying on the reductionist measures of quality, such as using the number of assaults as a proxy for safety, adding that there is a “rejection of staff and prisoners of the accuracy of the world painted by the data” [34] (p. 360). The MQPL+ is grounded in the daily experiences of people living and working within the environment. Quantitative analyses enable the benchmarking of the moral, relational, and emotional performances, and are sensitive to change. Liebling rejects the validity of capturing these qualities of institutional care uniquely through the use of brief scales. While limited in that these assessments require a high demand on time and human resources, the advances Liebling has made in this field offers important interdisciplinary learning for future efforts to develop current methodologies to measure psychiatric inpatient experience. QNIC accreditation reviews and CQC inspections are important governance processes for assessing inpatient CAMHS quality at a single timepoint—these inspections could benefit from Liebling’s conceptual understanding of social, moral, and relational experiences in addition to the robust methodology she has established. Her work makes a clear break from efforts to simply capture ‘satisfaction’. Measures of ‘therapeutic milieu’ have found important outcomes when comparing the value of the construct of social climate with satisfaction, and these ideas could bridge Liebing’s work with the application to inpatient CAMHS. 

### Measuring the Social Climates of Psychiatric Inpatient Settings

The essential ingredients of a therapeutic milieu—attachment, containment, communication, involvement, and agency [37]—have clear overlap with both Liebling’s work as well as the factors service users report as key to experience of psychiatric inpatient care. Banks and Priebe’s (2020) systematic review of instruments assessing therapeutic milieu in adult environments found five scales, four of which were developed between 1964–1984, and one in 2008 [38]. Of these scales, the Ward Atmosphere Scale (WAS) is the most widely used, and is psychometrically robust, but has 100 items, thereby limiting its use in routine practice. An adapted condensed version has been evaluated for assessing the ‘social climate’ in forensic inpatient settings [39]; this has evolved into the Essen Climate Evaluation Schema, with norms developed across 79 forensic psychiatric hospitals (EssenCES) [40,41]. 

The EssenCES has 15 items, measuring three dimensions of social climate: therapeutic hold, patients’ cohesion, and experienced safety. Siess and Schalast (2017) assessed the EssenCES for use in general adult psychiatric wards, finding good suitability for its use as a measure of social climate [41]. More recently, Efkemann, Bernard, and Kalagi et al. (2019) compared the experiences of ward atmosphere (EssenCES) and patient satisfaction for legally detained patients across psychiatric inpatient wards with contrasting locked door policies [42], a restrictive practice which is presently a fundamental dilemma for all psychiatric inpatient settings. In a mixed method study across four large acute psychiatric hospitals (average 253 beds) in Germany, each with different approaches to managing open door policies for involuntarily admitted patients, patients completed the EssenCES, along with an 8-item satisfaction scale ZUF-8 [43]), controlling for clinician-rated patient functioning. They found that better ward atmosphere—higher patient reported experienced safety and therapeutic hold—was associated with less restrictive door policies, as well as qualitative advantages to an open setting. In contrast, the authors found no significant differences according to a patient satisfaction questionnaire. Furthermore, there was little correlation between patient satisfaction and ward atmosphere—the authors concluded that these are likely independent factors. These findings highlight the value of examining the ward atmosphere in comparing inpatient experience across these settings. Once more the conclusions are based on a measure of inpatient satisfaction, the Zuf-8, which has not been developed for adolescent inpatient settings. The authors add that in a psychiatric inpatient environment the EssenCES has “face validity and clear relevance”, while in contrast “there is a need for an appropriate and more specific instrument to assess the patient satisfaction” [42] (p. 9).

Persistence in examining the ‘satisfaction’ of inpatient services may be misguided. There is a strong conceptual framework and research basis for focusing future efforts on the measurement of social climate, and the moral and relational performances of the inpatient CAMHS settings as a measure of adolescent inpatient experience. Such measures directly examine the factors that matter to service users in their inpatient experience. The association between the social climate of adolescent inpatient settings and the incidents occurring during these admissions, as well as post-discharge outcomes, remains unexamined. Liebling’s findings that institutional experiences are predictive on self-harm, suicide, violence, and personal growth provide an important signpost for future research within the inpatient CAMHS context. 

## 9. Conclusions 

CAMHS research and practice has been at the forefront of ensuring service user experience, and feedback is central to improving routine practice and shaping services. Inpatient CAMHS continues to lag behind. This is of critical importance, and there are currently no adequate tools to assess inpatient CAMHS experience. Measures lack face validity and fail to examine the factors we know that influence admission experience. In the context of 24/7 institutional care, power imbalances, acute distress, separation from family, restrictive practices, and elevated risks, ensuring that young people’s perspective on the care they receive should be at the forefront of appraising and improving services. Focus on reduced admissions and length of stay may have implicitly turned the attention away from the quality of inpatient experience, yet it is likely to be the key driver of improved overall outcomes. 

There are unique relational and technical challenges that remain. The current paper sets out an argument for moving away from assessing the ‘satisfaction with care’, towards the development of a measure of an inpatient CAMHS experience which examines the social climate, and the moral and relational quality of these settings. 

Future work to develop a measure of inpatient experience will need to draw on the ideas outlined above in collaboration with both adolescents and parents/carers who have experience of inpatient care. The current paper has not explored in depth the views of parents and carers, but their perspectives will be key in establishing a reliable and valid measure of inpatient experience. 

To be able to measure the quality of these services from the perspective of people with experience of the care seems fundamental to any possibility for progress. Prioritising young people’s moral, relational, and emotional experiences on CAMHS inpatient units is imperative, as to be treated with dignity and fairness is a matter of life and death [33]. 

## Data Availability

Not applicable.

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
