# Peer review of "Measuring Experience of Inpatient Child and Adolescent Mental Health Services (CAMHS)"

_ijerph, 2023, doi:10.3390/ijerph20115940_

Round 1

Reviewer 1 Report

Congratulations on writing on a subject that is rarely disputed!

Unfortunately, psychiatric problems tend to be neglected in many countries and often children grow up as adults with well-established behavioral issues and difficulties in having a normal life due to psychiatric underdiagnosed problems.

Your paper sheds some light on how young patients perceive their psychological guidance and psychiatric treatment.

But in my opinion, you should offer some quantifiable indicators about the feedback of patience. Maybe a score for each issue you are addressing would consolidate the scientific soundness of your paper. 

Your paper needs to be improved in order to publish it!

Author Response

Thank you for your feedback.  I have made multiple significant changes to address the concern regarding quantifiable indicatorsIn the opening sections, where possible I have tried to refer to strongest evidence including systematic reviews, and large-scale evidence NIHR reviews, with no meta-analyses available within this field.   Reference to qualitative research was used to provide the voice of the young person, this is in place of any large quantifiable data on the issue – I have made this point more explicit now in paragraph three of the section titled “Experiences of inpatient CAMHS”.  Then to better address the gap in terms of methodology and data I have firstly included more detail related to the ESQ validation studyI have also outlined explicitly in this section why my paper has been unable to cite any specific outcomes data using ESQ or other experience or satisfaction data in relevant settings, currently this is a gap in the evidenceFinally, I have expanded on the research methodology, design and outcomes from Liebling’s work as well as the use of EssenCES.    

Reviewer 2 Report

Congratulation on the interesting writing. A small suggestion, since this opinion piece is very much focusing on the United Kingdom background, perhaps to improve the title by adding "United Kingdom" or "NHS study" as the findings might not be applicable to other parts of the world / continent. Thank you  

Author Response

Thank you for your feedback 

Point 1 – inclusion of NHS or UK as findings may not be applicable in other parts of the world 

I agree that the context needs to be made clear, I would however like to avoid putting that in the title as I would like the ideas and application to be accessible and translatable across contexts.  I also want to keep the title as concise as possible, and I imagine that the acronym of CAMHS may alert some readers to the national context.  I have however added “in the UK” to the opening line of the abstract and it is also in the opening line of the introduction.  I hope that makes it clear that the paper is written from a UK context.   

Reviewer 3 Report

The article effectively draws from interdisciplinary sources to provide novel suggestions for gauging inpatient CAMHS experience, such as Liebling's work on moral and relational climates in prisons.

A thorough analysis of the literature is included, demonstrating solid awareness of the status of the research on the topic.

The article's dedication to enhancing inpatient CAMHS services is shown by its emphasis on the understanding and measuring of patient experiences as well as the significance of taking into account moral, relational, and emotional components of care.

The article's critical study of current metrics and tools strengthens the argument for creating new measurement methodologies, identifying their shortcomings in accurately representing inpatient CAMHS experiences.

Practical implications: The recommendations for developing new measures of inpatient CAMHS experience have clear, practical implications for the field, which could improve the quality of care and patient outcomes.

Recommendations:

Introduction: Provide the context for the present gap in understanding and quantifying the inpatient CAMHS experience. Provide a more detailed introduction to the ensuing parts.

Research methodology and design: Provide a more thorough explanation of the research procedures and designs employed in the cited papers. This will help readers comprehend the methodology behind these studies and the validity of their conclusions.

Presentation and structure: Organize the content into more precisely defined sections and subsections to enhance the overall presentation. Readers will find it simpler to follow the development of ideas and arguments.

Language and phrasing: Discuss the minimal English language changes that should be made throughout the text to increase cohesion and readability. Make sure sentences are concise, clear, and grammatically correct.

Other: Explain the importance of parents and caregivers in understanding and measuring the inpatient CAMHS experience, as their input is essential for creating a thorough evaluation instrument.

Author Response

Thank you for your supportive and constructive comments in reviewing our submission.  Below I will outline how we have addressed each point and made use of the comments to improve our paper.  A word document with track changes will allow reviewers to look over our proposed revisions alongside this cover letter.   

Point 2 – Provide more context for the present gap in understanding, more detailed introduction.  

Significant changes have been made to the introduction to attempt to signpost the reader to the thread of the arguments in the paper.   

Point 3 – Explain the importance of parents and caregivers in understanding and measuring inpatient CAMHS experience.   

I agree that it has been difficult to incorporate the multiple challenges related to capturing the experience of the young person on the ward and that of the parent/carer.  I have included a sentence in the introduction to clarify that purpose of the current paper is to examine the experience of the young person admitted.   

On pages 4 and 5 I have now provided more information about how studies and services have considered the balance between young people’s feedback and that of parents/ carers.  I have also now addressed more directly the difference between community CAMHS and inpatient CAMHS following the influential ESQ paper by Brown et al (2014) concluding that parent/carer feedback was more reliable.  In the conclusion I ensure that the importance of this consideration is held in mind for any future work.  

Point 4 – More precisely defined sections and subsections to enhance overall presentation 

I have not created new subsections.  I have deliberated on how to do so but have not been able to find a new solution.  I have reworded two of the subtitles, but I was unclear otherwise how to improve the reader’s experience in the development of the argument.  I hope that the additional information related to the research, as outlined in the following paragraph, may help the reader follow the arch of the argument we are making.  

Point 5 – Research methodology and design: Provide a more thorough explanation of the research procedures and designs employed in the cited papers. This will help readers comprehend the methodology behind these studies and the validity of their conclusions.

I have made multiple significant changes to address this concern.  In the opening sections, where possible I have tried to refer to strongest evidence including systematic reviews, and large-scale evidence NIHR reviews, with no meta-analyses available within this field.   Reference to qualitative research was used to provide the voice of the young person, this is in place of any large quantifiable data on the issue – I have made this point more explicit now in paragraph three of the section titled “Experiences of inpatient CAMHS”.  Then to better address the gap in terms of methodology and data I have firstly included more detail related to the ESQ validation study.  I have also outlined explicitly in this section why my paper has been unable to cite any specific outcomes data using ESQ or other experience or satisfaction data in relevant settings, currently this is a gap in the evidence.  Finally, I have expanded on the research methodology, design and outcomes from Liebling’s work as well as the use of EssenCES.    

Round 2

Reviewer 1 Report

There is a serious improvement in the initial article. The authors tried to answer to each and every request. The final paper is much better than the first one.

I really appreciate the work they've done.

A more thorough grammar correction should be done. But nevertheless, this paper has all it needs to be published in the present form.

Reviewer 3 Report

Congratulations. The changes made have increased the quality of the review.

Please address some minor spelling errors (e.g. "Yet there is a significant gap in evidence and practice").